# Unique membrane properties and enhanced signal processing in human neocortical neurons

Guy Eyal[1], Matthijs B Verhoog[2], Guilherme Testa-Silva[2], Yair Deitcher[3], Johannes C Lodder[2], Ruth Benavides-Piccione[4,5], Juan Morales[6], Javier DeFelipe[4,5], Christiaan PJ de Kock[2], Huibert D Mansvelder[2], Idan Segev[1,3]*

[1]Department of Neurobiology, The Hebrew University of Jerusalem, Jerusalem, Israel; [2]Department of Integrative Neurophysiology, Centre for Neurogenomics and Cognitive Research, VU University Amsterdam, Amsterdam, The Netherlands; [3]Edmond and Lily Safra Center for Brain Sciences, The Hebrew University of Jerusalem, Jerusalem, Israel; [4]Instituto Cajal, Madrid, Spain; [5]Laboratorio Cajal de Circuitos Corticales, Universidad Politécnica de Madrid, Madrid, Spain; [6]Escuela Técnica Superior de Ingenieros Informáticos, Universidad Politécnica de Madrid, Madrid, Spain

**Abstract** The advanced cognitive capabilities of the human brain are often attributed to our recently evolved neocortex. However, it is not known whether the basic building blocks of the human neocortex, the pyramidal neurons, possess unique biophysical properties that might impact on cortical computations. Here we show that layer 2/3 pyramidal neurons from human temporal cortex (HL2/3 PCs) have a specific membrane capacitance ($C_m$) of ~0.5 $\mu F/cm^2$, half of the commonly accepted 'universal' value (~1 $\mu F/cm^2$) for biological membranes. This finding was predicted by fitting in vitro voltage transients to theoretical transients then validated by direct measurement of $C_m$ in nucleated patch experiments. Models of 3D reconstructed HL2/3 PCs demonstrated that such low $C_m$ value significantly enhances both synaptic charge-transfer from dendrites to soma and spike propagation along the axon. This is the first demonstration that human cortical neurons have distinctive membrane properties, suggesting important implications for signal processing in human neocortex.

*For correspondence: idan@lobster.ls.huji.ac.il

**Competing interests:** The authors declare that no competing interests exist.

## Introduction

Since the beginnings of modern neuroscience the neocortex has attracted special attention because it is considered to play a key role in human cognition. Starting with the seminal work of Santiago Ramón y Cajal (*Cajal, 1995*) and Camilo Golgi (*Golgi, 1906*), and continuing with modern anatomical studies (*Jacobs et al., 2001*; *Watson et al., 2006*; *DeFelipe, 2011*), anatomists have been fascinated by its cellular structure. Comparison of human and rodents cortices shows that the human cortex is thicker (in particular layer 2/3 [*DeFelipe et al., 2002*; *Elston et al., 2001*]), contains more white matter (*Herculano-Houzel et al., 2010*), its neurons are larger (*Mohan et al., 2015*), and its cortical pyramidal cells have more synaptic connections per cell (15,000–30,000 for layer 2/3 pyramidal neurons [*DeFelipe, 2011*; *DeFelipe et al., 2002*]). It seems that the human neocortex, and especially its neurons, is anatomically unique. But what about the biophysical properties, e.g. the specific membrane capacitance of human cortical neurons and their cable characteristics? These have been shown to be key in determining information processing in neurons and in the networks they form (*Segev and Rall, 1988*; *Major et al., 1994*; *Stuart and Spruston, 1998*; *Magee, 1998*; *Segev and*

*London, 2000*; *Poirazi and Mel, 2001*; *London and Häusser, 2005*; *Hay et al., 2011*; *Eyal et al., 2014*; *Markram et al., 2015*).

To probe the biophysical properties of cortical neurons we built detailed 3D cable and compartmental models of L2/3 pyramidal cells from the human temporal cortex (HL2/3 PCs). The models were based on in vitro intracellular physiological and anatomical data from these same cells. To collect this data, fresh cortical tissue was obtained from brain operations at the neurosurgical department in Amsterdam. Details can be found in our previous works (*Testa-Silva et al., 2010*; *Verhoog et al., 2013*; *Testa-Silva et al., 2014*; *Mohan et al., 2015*), as well as in the work of other groups (*Schwartzkroin and Prince, 1978*; *Szabadics et al., 2006*; *Köhling and Avoli, 2006*). Our models of human neurons also incorporate data on dendritic spines obtained from light-microscope studies in HL2/3 PCs (*Benavides-Piccione et al., 2013*; *DeFelipe et al., 2002*; *Elston et al., 2001*). These are the first-ever detailed models of human neurons. All modeled cells, implemented in NEURON (*Carnevale and Hines, 2006*), and the corresponding anatomical and physiological data can be found in *Source code 1* and in ModelDB (https://senselab.med.yale.edu/ModelDB/showmodel. cshtml?model=195667).

## Results

In *Figure 1a*, fifty brief depolarizing current pulses (200 pA, 2 ms each) were injected into the soma of the HL2/3 PC shown at the right; the resultant averaged voltage transient is depicted by the black trace. That same cell was then reconstructed in 3D and the dendritic spines were added globally as in *Rapp et al. (1992)*, see Materials and methods. The shape, size, membrane area, and distribution of dendritic spines in the HL2/3 model were based on high-resolution 3D confocal images of several stretches of HL2/3 PCs dendrites (See Materials and methods, Figure 4 and more details in *Benavides-Piccione et al., 2013*). The 3D anatomy plus physiology were combined for reconstructing a detailed model for that cell. Attempts to fit the experimental transient via the model are depicted by the various colored theoretical transients (see Materials and methods), each with its corresponding values for axial resistivity, $R_a$, specific membrane resistivity, $R_m$, and capacitance, $C_m$. The green transient (with $R_a$ = 203 Ωcm, $R_m$ = 38,907 Ωcm² and $C_m$ = 0.45 µF/cm²) best fitted the experimental transient. Attempts to fit the transient with larger values of $C_m$ failed (red and magenta traces). A close fit between model and experimental transients, with identical cable parameters as the best model in *Figure 1a*, was obtained for a range of (depolarizing and hyperpolarizing) voltage transients recorded experimentally from that same cell (*Figure 1b*). A similar range of cable parameters was also found for five additional modeled and experimentally characterized HL2/3 pyramidal cells (*Figure 1c1–c5*). Furthermore, the low value for $C_m$, around 0.5 µF/cm², was found in all these five additional modeled cells ($C_m$= 0.47 ± 0.03 µF/cm², mean ± S.D, n = 6). A summary of the results in *Figure 1* is provided in *Figure 1—figure supplement 1a1–a3*.

To validate that low $C_m$ values are indeed necessary to achieve good fits between the theoretical and the experimental transients, we plotted (*Figure 1—figure supplement 1b–d*) the root mean square deviation (RMSD) for different combinations of the three passive parameters ($C_m$, $R_m$, $R_a$) that impact the transients. In all cases, the RMSD obtained with $C_m$ = 0.5 µF/cm² is much smaller than the one obtained with the larger $C_m$ value (*Figure 1—figure supplement 1b*). $R_m$ values are cell-specific (*Figure 1—figure supplement 1c*), whereas the value of $R_a$ has a negligible impact on the RMSD (*Figure 1—figure supplement 1d*). As in (*Major et al., 1994*) we found that large $R_a$ value improves the fit over the first two milliseconds of the transient (not shown). Indeed, the best way to directly measure $R_a$ is to use two (somatic and dendritic) electrodes, but this was beyond the scope of the present study. The values in *Figure 1* for $R_a$ (268.5 ± 30.0 Ωcm, n = 6) should therefore be viewed with some caution. However, we would like to re-emphasize that our prediction of low $C_m$ value in human L2/3 pyramidal neurons holds independently of the $R_a$ value chosen (*Figure 1— figure supplement 1b*). We further examined whether possible statistical and systematic errors in the data might effect our estimates of $C_m$, $R_m$ and $R_a$ by performing error analysis as in *Roth and Häusser (2001)*. We found that even with large systematic errors in morphological parameters, the best fit still yields low $C_m$ value in human L2/3 pyramidal neurons (see *Table 1*).

The low $C_m$ value ~ 0.5 µF/cm² found in HL2/3 PCs is surprising; it is commonly assumed that $C_m$ is close to 1 µF/cm² (*Hodgkin et al., 1952*; *Cole, 1968*), as was modeled by many researchers on rodent neurons (*Major et al., 1994*; *Roth and Häusser, 2001*; *Nörenberg et al., 2010*;

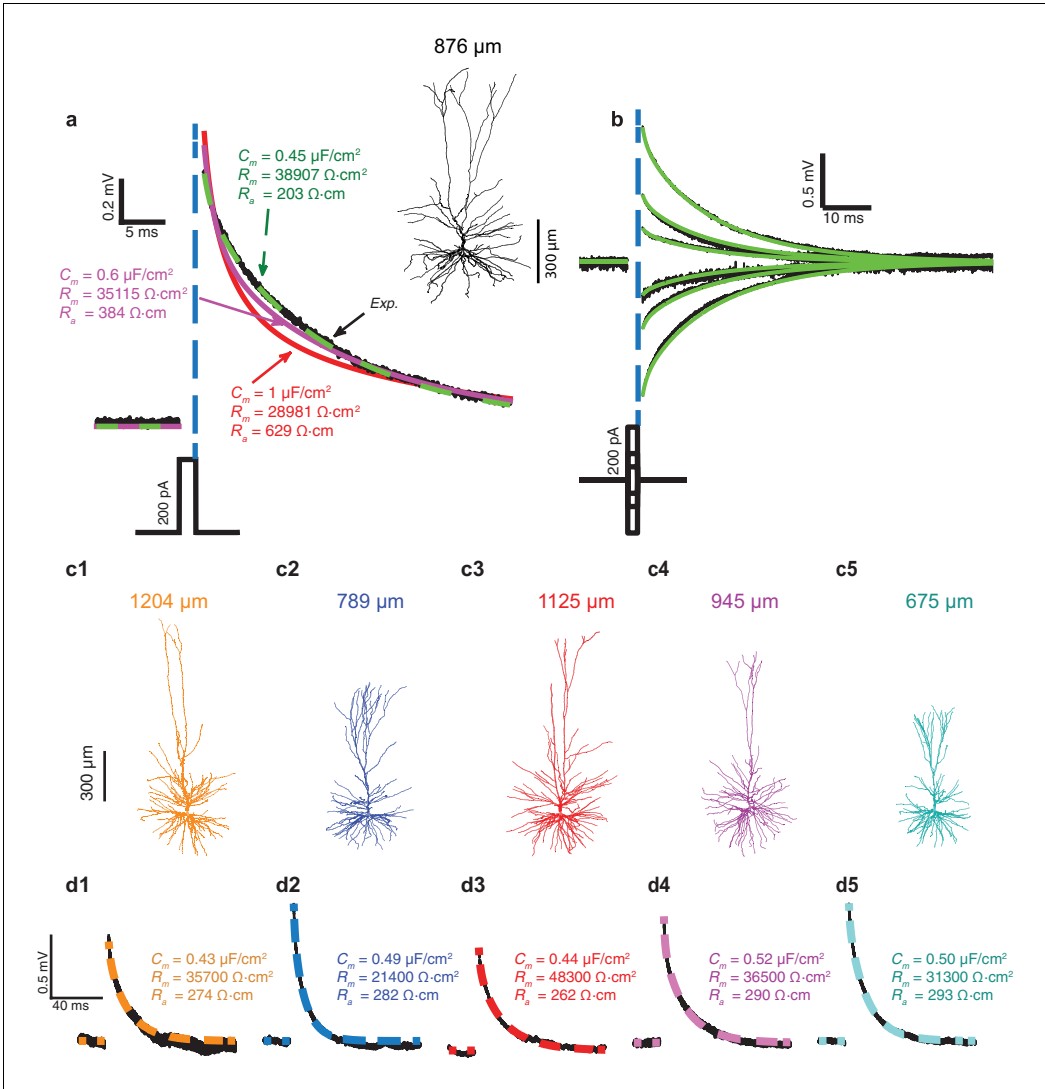

**Figure 1.** Exceptionally low specific membrane capacitance, $C_m$, for L2/3 human pyramidal cells predicted using detailed neuron models. (a) Experimental transient voltage response (upper black trace, *Exp.*) to a brief somatic depolarizing current pulse (2 ms, 200 pA). The corresponding 3D reconstructed HL2/3 cell from which this voltage transient was obtained is shown in the inset. Color traces depict theoretical transients resulting from injecting the experimental current into the compartmental model of that cell (see Materials and methods). The corresponding cable parameters ($C_m$, $R_m$, $R_a$) are shown for each theoretical transient. Excellent fit was obtained only with $C_m$ value near 0.5 µF/cm² (green trace), which is half the conventional value of around 1 µF/cm², obtained for a variety of neuron types, including cortical neurons of other species. (b) Model parameters with low $C_m$ (green traces in a) also match the experimental responses (black) to stimuli of different strength and signs (−200 pA, −100 pA, −50 pA, 50 pA, 100 pA, 200 pA). (c1–c5) Five additional human L2/3 neurons taken from our human neuron database (depth from pia is provided above each cell). (d1–d5) The experimental transient voltage response for the neurons in c1–c5 (black traces), and models fit to the transients (overlapping color traces). In all models, $C_m$ ranged between 0.43 µF/cm² and 0.52 µF/cm². Patient history for these six cells is described in *Table 2*.

The following figure supplements are available for figure 1:

**Figure supplement 1.** Estimates for passive properties of human L2/3 neurons.

**Figure supplement 2.** Cable properties for mouse L2/3 pyramidal cells from the temporal cortex.

**Figure supplement 3.** Estimating passive parameters for human L2/3 pyramidal cells when incorporating Ih channels into the model.

**Table 1.** Estimated statistical and systematic errors in the six human L2/3 PCs models shown in *Figure 1* as in (*Roth and Häusser, 2001*). The mean relative statistical error was 6.5% in $C_m$ (range, 4.2-9.2%, n = 5), 8.6% in $R_m$ (range, 4.2-18.8%, n = 5), and 12.8% in $R_a$ (range, 5.3-20.2%, n = 5). In cell #060311, larger errors were found due to several noisy traces. These range of errors in the estimated values of $C_m$, $R_m$ and $R_a$, are within the range of values found for the six modeled cells in *Figure 1*. We also analyzed the case of systematic errors; the errors introduced are shown in *Table 3*. This leads to, 40% mean relative error in $C_m$ (range, 36.7–58.2%, n = 6), 30.1% in $R_m$ (range, 28.6–34.1%, n = 6), and 53.0% in $R_a$ (range, 47.3–65.4%, n = 6). Thus, even large systematic errors in morphological parameters cannot explain the low $C_m$ (~0.5 µF/cm$^2$) in human L2/3 pyramidal neurons as compared to the typical value ~1 µF/cm$^2$ found in other neurons.

| | Cell 060303 (*Figure 1c2*, blue) | | | Cell 060308 (*Figure 1a*, green) | | | Cell 060311 (*Figure 1c3*, red) | | |
|---|---|---|---|---|---|---|---|---|---|
| | Best fit | S.D. stat | S.D. sys | Best fit | S.D. stat | S.D. sys | Best fit | S.D. stat | S.D. sys |
| $C_m$ (µF/cm$^2$) | 0.488 | 0.045 | 0.183 | 0.452 | 0.026 | 0.196 | 0.441 | 0.248 | 0.257 |
| $R_m$ (kΩ•cm$^2$) | 21.41 | 1.21 | 6.57 | 38.91 | 1.65 | 13.29 | 48.73 | 22.90 | 14.96 |
| $R_a$ (Ω•cm) | 281.78 | 34.12 | 152.05 | 203.23 | 29.31 | 96.18 | 261.97 | 179.3 | 127.22 |
| | Cell 130303 (*Figure 1c1*, orange) | | | Cell 130305 (*Figure 1c5*, cyan) | | | Cell 130306 (*Figure 1c4*, magenta) | | |
| | Best fit | S.D. stat | S.D. sys | Best fit | S.D. stat | S.D. sys | Best fit | S.D. stat | S.D. sys |
| $C_m$ (µF/cm$^2$) | 0.430 | 0.034 | 0.158 | 0.497 | 0.027 | 0.207 | 0.520 | 0.0219 | 0.236 |
| $R_m$ (kΩ•cm$^2$) | 35.67 | 7.40 | 11.62 | 31.31 | 1.99 | 8.96 | 36.52 | 2.89 | 11.51 |
| $R_a$ (Ω•cm) | 274.44 | 32.24 | 171.63 | 292.95 | 59.04 | 164.46 | 290.28 | 15.41 | 137.46 |

*Szoboszlay et al., 2016*) and directly confirmed for several classes of neurons using patch recordings from nucleated membrane patches (*Gentet et al., 2000*). For self-consistency, we repeated the same experimental and theoretical protocols as in *Figure 1* on L2/3 PCs from the mouse temporal cortex (n = 4). In contrast to human neurons, and in agreement with the existing literature, we found that the best theoretical fit to the experimental transients yields $C_m$ of 1.09 ± 0.36 µF/cm$^2$ (*Figure 1—figure supplement 2*). This value is significantly larger compared to that found in *Figure 1* for human L2/3 neurons (students t-test without the assumption of equal variance, p-val = 0.0408, non parametric Mann-Whitney U-test, p-val = 0.0095). It is important to note that the low value of $C_m$ in human neurons is associated with a high $R_m$, such that the membrane time constant ($\tau_m = C_m*R_m$) is in the range of 10–22 ms ($\tau_m$ = 16.5 ± 3.7, n = 6), which is similar to the value found in the mouse L2/3 pyramidal neurons ($\tau_m$ = 16.1 ± 1.3, n = 4) and in rat L2/3 pyramidal cells ($\tau_m$ = 13.1 ± 1.7, [*Sarid et al., 2007*]). This implies that the temporal resolution (the synaptic integration time-window) of human and rodents' pyramidal L2/3 cells is similar.

Could it be that our uniform passive model used in *Figure 1* is too simplistic? Perhaps non-uniform distribution of $R_m$ in the dendrites (as was shown in rodent layer 5 PCs [*Stuart and Spruston, 1998*] and in inhibitory basket cells [*Nörenberg et al., 2010*]), or active membrane currents such as Ih that are open around the resting membrane potential (*Kole et al., 2006*; *Harnett et al., 2015*), affect our estimation for $C_m$? These two possibilities were examined via simulations. In the first set of simulations $R_m$ was assumed to be spatially non-uniform in the dendrites of the six modeled cells shown in *Figure 1*. In each case, the cable properties were optimized in order to best fit the experimental transients (see details in the Materials and methods). In all six cases the estimated $C_m$ for the optimal non-uniform case was similar to the value obtained when $R_m$ was uniform (not shown).

In the next set of simulations Ih was added to the model; the density of the Ih current increased with distance from soma as in *Kole et al. (2006)*, see Materials and methods. The cable properties for the six modeled cells shown in *Figure 1* were optimized in order to generate the closest fit between model and experimental transients (see details in the Materials and methods). We found that the larger the Ih density, the larger was the estimated $C_m$ value that best fitted the experimental transient (from ~0.45 µF/cm$^2$ without Ih, *Figure 1—figure supplement 3a1*, to ~0.76 µF/cm$^2$ when Ih density at the soma was 0.2 mS/cm2, as found in rat L5PC (*Kole et al., 2006*), *Figure 1—figure supplement 3c1*). However, the quality of the fit was significantly reduced compared to the passive case particularly at short times (*Figure 1—figure supplement 3b2,c2*). Indeed, in the first few

milliseconds, the voltage transient is dominated by the capacitive membrane current (and not by the ionic currents, e.g., Ih); at these short times, a close match between model and experiments was obtained only with $C_m \sim 0.5$ µF/cm$^2$, even in the presence of Ih (*Figure 1—figure supplement 3d1*). We therefore conclude that in order to explain the experimental transients in human L2/3 neurons, $C_m$ should be ~0.5 µF/cm$^2$.

This puzzling result of low $C_m$, values in human neurons inspired us to perform a new set of experiments, using nucleated patches from HL2/3 PCs, in order to directly measure $C_m$, exactly as in *Gentet et al. (2000)*. We obtained an experimental value for $C_m$ of 0.51 ± 0.11 µF/cm$^2$ (mean ± S.D, n = 5 neurons, see details on these cells in *Table 2*), in full agreement with our model prediction (*Figure 2*). Repeating the same procedure in mouse neurons resulted in a $C_m$ of 0.83 ± 0.34 µF/cm$^2$ (n = 7 neurons from 7 mice; *Figure 2d*), similar to the findings of (*Gentet et al., 2000*; *Szoboszlay et al., 2016*) and significantly different from the human $C_m$ values (students t-test following the KS test for normality (*Massey, 1951*), p-val = 0.0472). Combining the $C_m$ values obtained in both methods, the model fittings (*Figure 1*) and the nucleated patches (*Figure 2*) shows the large difference in $C_m$ between human ($C_m$ = 0.49 ± 0.08 µF/cm$^2$, n = 11) and mouse ($C_m$ = 0.93 ± 0.36 µF/cm$^2$, n = 11). This difference is highly significant (*Figure 2d*, students t-test following the KS test for normality, p-val = 0.0021, non-parametric Mann-Whitney U-test, p-val = 0.006). Thus, we find that human L2/3 pyramidal neurons have different membrane capacitance compared to that of rodent pyramidal neurons.

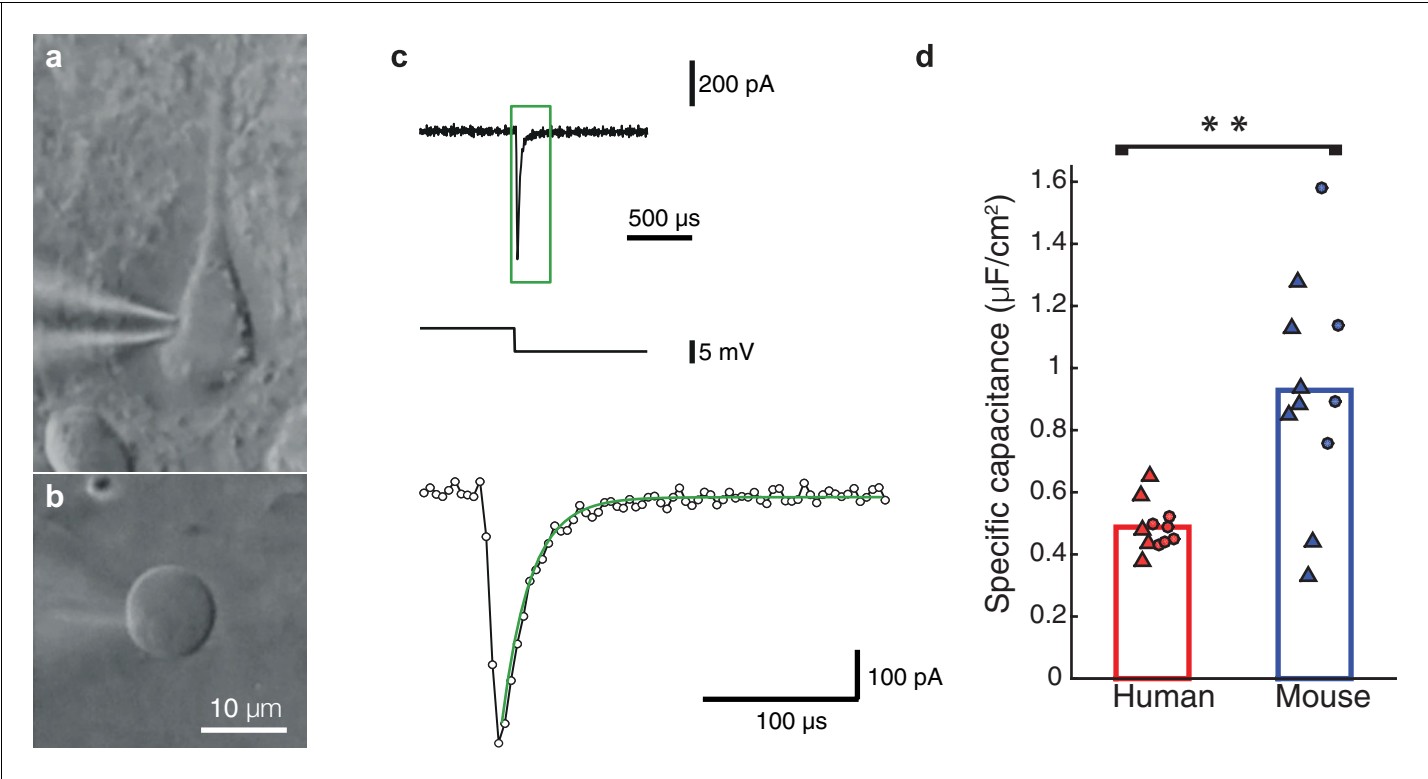

**Figure 2.** Nucleated-patch measurement of the specific membrane capacitance in HL2/3 neurons directly validated model prediction for the low $C_m$ value in these cells. (a) Differential Interference Contrast (DIC) image of a human L2/3 pyramidal neuron in an acute brain slice of human temporal cortex targeted for whole-cell patch clamp recording. (b) Nucleated soma patch pulled from the neuron in a. (c) Top trace: Current response of the nucleated patch shown in a to a −5 mV hyperpolarizing step (middle trace). Bottom trace: magnification of the green box in the top trace, showing mono-exponential fit (green line) to the capacitive current from which the total membrane capacitance was extracted. This capacitance was then divided by the surface area of the patch, providing the specific membrane capacitance (see Materials and methods). (d) Summary plot of the specific membrane capacitance from human and mouse. Triangles, $C_m$ values from the nucleated patches. Circles, $C_m$ values obtained from fitting model to experimental transients (as in *Figure 1* and *Figure 1—figure supplement 2*); human (red, n = 11, see details in *Table 2*), mouse (blue, n = 11). p-val = 0.0021 using students t-test following the KS test for normality.

To examine the impact of the low $C_m$ value found in HL2/3 neurons on signal processing in these cells, we used the detailed compartmental model for the cell shown in *Figure 1a*, with model parameters that fit the experimental transients (green curve and corresponding cable parameters). By activating excitatory spiny synapses at both the distal tuft and a distal basal dendrite, we first examined in *Figure 3a* the impact of the low $C_m$ value on the synaptic charge transfer in HL2/3 dendrites. We found that the peak somatic excitatory postsynaptic potential (EPSP) is larger by ~84% (basal synapse) to ~93% (apical synapse) when $C_m$ = 0.45 µF/cm$^2$ (red traces) as compared to the case with $C_m$ = 0.9 µF/cm$^2$ (blue traces). We also measured the dendritic delay, defined as the time-difference between that of the local peak EPSP to that of the resultant soma EPSP (*Agmon-Snir and Segev, 1993*). The dendritic delay from the distal tuft and basal synapses was dramatically reduced from 39 ms and 11 ms when $C_m$ = 0.9 µF/cm$^2$ to 20 ms and 6.5 ms respectively, when $C_m$ = 0.45 µF/cm$^2$ (see complementary *Figure 3—figure supplement 1*).

We next quantified how many excitatory spinous synapses should be simultaneously activated to initiate a somatic Na$^+$ spike in our model for HL2/3 PCs (see Materials and methods). We have used excitatory synapses with both AMPAR and NMDAR currents, with peak conductance values of 0.7 nS and 1.4 nS respectively (See Materials and methods). These synapses were randomly distributed over the modeled HL2/3 dendrites and simultaneously activated. With $C_m$ = 0.45 µF/cm$^2$, about 100 excitatory synapses were required for generating a somatic spike with 50% probability, whereas about 170 synapses were required with $C_m$ = 0.9 µF/cm$^2$. The lower $C_m$ value as found in HL2/3 pyramidal cells both improved synaptic charge transfer from dendrites to soma (larger EPSPs and reduced dendritic delay) and also reduced the number of synapses required to initiate a somatic/axonal spike, thus compensating for the increase in the size of the dendritic tree in human neocortex (*Mohan et al., 2015*).

We further explored the impact of $C_m$ ~ 0.5 µF/cm$^2$ on the velocity of spike propagation along the axon (*Figure 3c*). We examined the case of a long non-myelinated axon, assuming that $C_m$ in the axon is similar to that of the cell body. As expected, a lower $C_m$ value increased the propagation speed of the action potential; for the parameters we used, the propagation was about 65% faster compared with the case of a higher $C_m$. This is in agreement with the analytical result obtained by (*Jack et al., 1975*, page 432) for the impact of $C_m$ on spike propagation velocity. Thus, low $C_m$ value enables an efficient intra- and inter-regional information-transfer despite the large size of the human brain.

## Discussion

Our study provides the first direct evidence for the unique properties of the membrane in human cortical cells. In particular, we showed that layer 2/3 pyramidal neurons from human temporal cortex have a specific membrane capacitance ($C_m$) of ~0.5 µF/cm$^2$, half of the commonly accepted 'universal' value (~1 µF/cm$^2$) for biological membranes. It is important to note that a few studies have demonstrated that $C_m$ in rodent's pyramidal neurons may range between 0.7–2 µF/cm$^2$ (*Major et al., 1994*; *Thurbon et al., 1998*), but direct measurements using nucleated patches found a much narrower range, 0.8–1.1 µF/cm$^2$, in cortical pyramidal neurons (*Gentet et al., 2000*; *Szoboszlay et al., 2016*), as well as in cerebellar Golgi cells (*Szoboszlay et al., 2016*). This narrow range of values also agrees with the modeling prediction of (*Nörenberg et al., 2010*) for GABAergic hippocampal interneurons, where a correction for the spines area is not required. In the present work, we used both a modeling approach, as well as nucleated patch experiments, to demonstrate that $C_m$ in human L2/3 pyramidal neurons is significantly lower than in rodents. We next demonstrated that such low $C_m$ value has important functional implications for signal processing in dendrites and axons.

One may argue that the neurons we used have abnormal membrane properties as these neurons originated from humans with epileptic seizures. However, we only used brain samples that were not part of the diseased tissues, but had to be removed in order to gain access to deeper brain structures for surgical treatment. The eleven HL2/3 pyramidal cells used in this research (six cells in *Figure 1* and five cells in *Figure 2*) were taken from five different human patients with different disease records that were treated with different drugs (see *Table 2*). Yet, in all cases we found similar membrane properties. Furthermore, we recently have shown that there is no correlation between the disease history (epilepsy onset and total numbers of seizures) and the neuron morphology (*Mohan et al., 2015*).

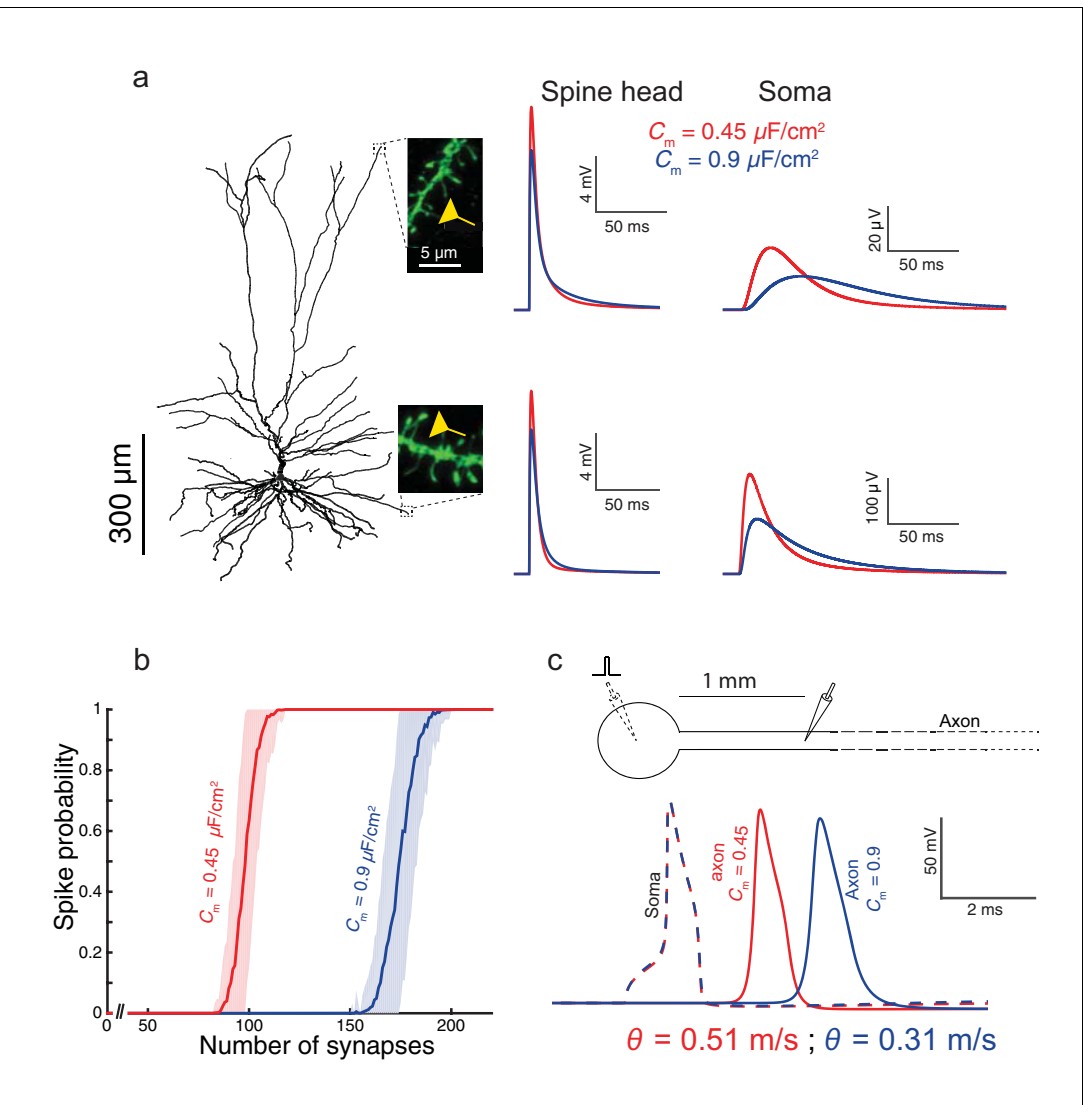

**Figure 3.** Functional implications of the low $C_m$ value in human L2/3 cortical neurons for signal processing. (a) The neuron model shown in *Figure 1a*, receiving an excitatory synaptic input on distal apical dendritic spine (top trace) and on a distal basal dendritic spine (lower traces). The model cell has either $C_m = 0.45$ µF/cm² (red traces) or $C_m = 0.9$ µF/cm² (blue traces), while the other cable parameters ($R_a = 203$ Ωcm, $R_m = 38,907$ Ωcm²) were kept fixed. Excitatory synapses were activated on the head of the modeled dendritic spine (inset, scale bar is 5 µm; see Materials and methods). Note the larger and faster somatic EPSP for the red case. (b) For the cell model shown in **a**, significantly smaller number of excitatory spinous synapses were required for initiation of a somatic spike when $C_m = 0.45$ µF/cm². Synapses were simultaneously activated and distributed randomly over the dendritic tree (see Materials and methods). (c) Top, schematics of soma and axon of the cell modeled in **a**., the axon had a diameter of 1 µm. Bottom, the velocity (θ) of the axonal spike, measured at 1 mm from the soma, is significantly increased (by about 65%) with $C_m = 0.45$ µF/cm² (red spike); the amplitude of the propagated axonal spike is also slightly increased in this case.

The following figure supplement is available for figure 3:

**Figure supplement 1.** The functional implications of the low $C_m$ value for the case where the membrane time constant, $\tau_m$, is kept fixed by a corresponding change in $R_m$ for both $C_m = 0.45$ µF/cm² and $C_m = 0.9$ µF/cm².

Unfortunately, we do not yet have a comprehensive comparative study for assessing the value of $C_m$ in other large brains, e.g., those of non-human primates. Recent compartmental models of pyramidal neurons in L3 of the prefrontal cortex of rhesus monkeys suggested that aged animals have

smaller $C_m$ values (~0.7 µF/cm$^2$) than young animals (~1.1 µF/cm$^2$) (*Rumbell et al., 2016*). Further direct measurements of $C_m$, e.g., using the nucleated patch technique in different animals, could provide insights to the evolution of the specific capacitance of neuronal membranes.

What could be the reason for the low $C_m$ as found in the present work? The specific capacitance is determined by the dielectric constant of the material composing the membrane and by the membrane thickness. We measured the membrane thickness in a few samples from human and mouse L2/3 pyramidal cells, both from the temporal cortex; both showed similar thickness of 5.1–5.8 nm (not shown). Another possibility that might explain differences in $C_m$ is the compositions of neuronal membranes. *Gentet et al. (2000)* demonstrated that transmembrane proteins have small effect on $C_m$. However, a recent study showed that $C_m$ of condensed phosphatidylcholine-based monolayers is sensitive to their lipid composition and molecular arrangement (*Lecompte et al., 2015*). Furthermore, *Szoboszlay et al. (2016)*, using nucleated patch experiments on mouse cerebellar neurons, showed that $C_m$ increases significantly (from 1 µF/cm$^2$ to more than 2 µF/cm$^2$) after adding Mefloquine to the bath; Mefloquine is known to bind to the membrane phospholipids (*Chevli and Fitch, 1982*). Interestingly, (*Bozek et al., 2015*) showed a significant difference in the lipidome of human, chimpanzee, macaque, and mouse. More specifically, *Moschetta et al. (2005)* and *Chan et al. (2012)* showed differences in the ratio between phospholipids, cholesterol, and sphingomyelin in human vs. rodents. In view of the above studies, we suggest that the lipid composition of HL2/3 membrane explains their low $C_m$. This assertion requires direct examination of the human membranes at the molecular level, which is beyond the scope of the present study.

Whatever the reason, the low $C_m$ effectively enhances signal transfer (both synaptic and action potentials, *Figure 3* and *Figure 3—figure supplement 1*) and increases the neuron's excitability, so that less depolarizing charge is required for generating Na$^+$ spikes and for supporting dendritic backpropagating Na$^+$ action potentials (*Stuart and Sakmann, 1994*). We conclude by suggesting that the distinctive biophysical membrane properties of pyramidal neurons in human evolved as an outcome of evolutionary pressure and resulted in an efficient information transfer, countering the increase in size/distances in the human brain.

## Materials and methods

### Experimental data

#### Layer 2/3 neurons from live human tissue

All procedures on human tissue were performed with the approval of the Medical Ethical Committee (METc) of the VU University Medical Centre (VUmc), with written informed consent by patients involved to use brain tissue removed for the treatment of their disease for scientific research, and in accordance with Dutch license procedures and the declaration of Helsinki (VUmc METc approval 'kenmerk 2012/362'). Slices of human temporal cortex were cut from neocortical tissue that had to be removed to enable the surgical treatment of deeper brain structures for epilepsy or tumors. Patients (32–43 years of age) provided written informed consent prior to surgery. In all patients, the resected neocortical tissue was located outside the epileptic focus or tumor, and displayed no structural/functional abnormalities in preoperative MRI investigations. After resection, the neocortical tissue was placed within 30 s in ice-cold artificial cerebrospinal fluid (aCSF) slicing solution which contained in (mM): 110 choline chloride, 26 NaHCO3, 10 D-glucose, 11.6 sodium ascorbate, 7 MgCl2, 3.1 sodium pyruvate, 2.5 KCl, 1.25 NaH2PO4, and 0.5 CaCl2 – 300 mOsm, saturated with carbogen gas (95% O2/5% CO2) and transported to the neurophysiology laboratory, which is located 500 meters from the operating room. The transition time between resection of the tissue and the start of preparing slices was less than 15 min. Neocortical slices (350–400 µm thickness) were prepared in ice-cold slicing solution, and were then transferred to holding chambers filled with aCSF containing (in mM): 126 NaCl; 3 KCl; 1 NaH2PO4; 1 MgSO4; 2 CaCl2; 26 NaHCO3; 10 glucose – 300 mOsm, bubbled with carbogen gas (95% O2/5% CO2). Here, slices were stored for 20 min at 34°C, and for at least 30 min at room temperature before recording. Whole-cell, patch clamp electrophysiology recordings were then made from human layer 2/3 pyramidal neurons as described previously (*Verhoog et al., 2013*; *Testa-Silva et al., 2014*). Whole-cell recording electrodes were uncoated and glass thickness was 0.64 mm. A multiclamp 700B (molecular devices) was used for the recordings. Recording aCSF was the same solution as the aCSF in which slices were stored.

Recording temperature was 32–35℃. Internal solutions were (in mM): 110 Kgluconate; 10 KCl; 10 HEPES; 10 K2Phosphocreatine; 4 ATP-Mg; 0.4 GTP, biocytin 5 mg/ml, pH adjusted with KOH to 7.3 (290–300 mOsm).

## Human neurons (postmortem)

Samples obtained from 2 human males (aged 40 and 85) were used in this study. This tissue (kindly supplied by Dr I. Ferrer, Instituto de Neuropatología, Servicio de Anatomía Patológica, IDIBELL-Hospital Universitario de Bellvitge, Barcelona, Spain) was obtained at autopsy (2–3 hr post-mortem). Brain samples were obtained following the guidelines and approval by the Institutional Ethical Committee. The cause of death was a traffic accident (case C40) and pneumonia plus interstitial pneumonitis (aged case, C85); these cases were used also in *Benavides-Piccione et al. (2013)*. Their brains were immediately immersed in cold 4% paraformaldehyde in 0.1 M phosphate buffer, pH 7.4 (PB) and sectioned into 1.5-cm-thick coronal slices. Small blocks of the cortex (15 × 10 × 10 mm) were then transferred to a second solution of 4% paraformaldehyde in PB for 24 hr at 4℃. In the present study, the tissues used were from the cingulate gyri and from the temporal cortex, corresponding to Brodmann's area 24 and 20, respectively (*Brodmann, 2007*). Coronal sections (250 µm) were obtained with a Vibratome and labeled with 4,6 diamino-2-phenylindole (DAPI; Sigma, St Louis, MO, United States of America) to identify cell bodies. Pyramidal cells were then individually injected with Lucifer yellow (LY; 8% in 0.1 M Tris buffer, pH 7.4), in cytoarchitectonically identified layer 3 cells. LY was applied to each injected cell by continuous current, until the distal tips of each cell fluoresced brightly, indicating that the dendrites were completely filled, and this ensured that the fluorescence did not diminish at a distance from the soma. Following the intracellular injection of pyramidal neurons, sections were processed with a rabbit antibody against LY produced at the Cajal Institute (1:400,000 in stock solution containing 2% BSA (A3425, Sigma), 1% Triton X-100 (30632, BDH Chemicals), 5% sucrose in PB). Antibody binding was detected with a biotinylated donkey anti-rabbit secondary antibody (1:100 in stock solution; RPN1004, Amersham, Buckinghamshire, United Kingdom) followed by a solution of streptavidin coupled to Alexa Fluor 488 (1:1000; Molecular Probes, Eugene, OR, USA). Finally, sections were mounted in 50% glycerol in PB.

## Mouse

All animal experimental procedures were approved by the VU University's Animal Experimentation Ethics Committee and were in accordance with institutional and Dutch license procedures (approved protocol INF09-02A1V1). Coronal slices (350–400 µm thickness) were cut from the temporal association cortex of C57Bl6 mice (2–3 weeks of age). As in the preparation of human brain slices, slices were allowed to recover for 30 min at 34°C followed by 30 min at room temperature. Solutions for human and mouse slice recordings were identical.

## 3D reconstructions of HL2/3 pyramidal cells and of mouse L2/3 pyramidal cells (acute living slices)

Six morphologies of human L2/3 cells, residing at the depths of 675–1204 µm below the pia, and four morphologies of mouse L2/3 cells from depths of 224–378 µm, were reconstructed in 3D via Neurolucida software (Microbrightfield, Williston, VT, USA), using a 100x oil objective. The human morphologies appear in the database from *Mohan et al. (2015)*. Additional details regarding the reconstruction methods can be found there.

## Reconstructing dendritic spines (postmortem)

Dendritic spines and dendritic branches were imaged using a Leica TCS 4D confocal scanning laser attached to a Leitz DMIRB fluorescence microscope. Consecutive stacks of images (3 ± 0.6 stacks per dendrite; 52 ± 17 images; z-step of 0.28 µm) were acquired using a 0.075 × 0.075 × 0.28 µm³ voxel size (Leica Objective Plan-Apochromat 63x/1.30 NA glycerol DIC M27) to capture the full dendritic depths, lengths, and widths (*Figure 4*). Spine structure was analyzed in 40 basal dendrites and 32 main apical dendrites (10 basal dendrites and 8 main apical dendrites per case and per cortical area), using Imaris 6.4.0 (Bitplane AG, Zurich, Switzerland). Since there are no clear structural borders between the head and the neck of a spine, no such distinction was applied. Instead, we reconstructed the complete morphology of each dendritic spine in 3D. Correction factors used in other

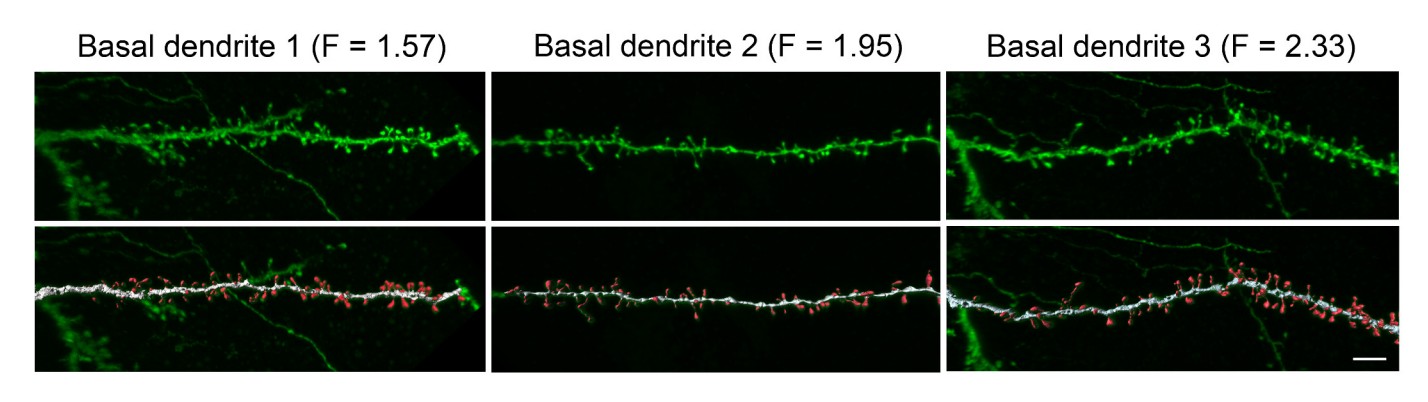

Basal dendrite 1 (F = 1.57)    Basal dendrite 2 (F = 1.95)    Basal dendrite 3 (F = 2.33)

**Figure 4.** High resolution reconstruction of spines from HL2/3 dendrites. Top. Three examples of confocal images of basal dendritic branches from post mortem tissue. Bottom. The 3D images were segregated to spines (red) and shafts (white) that were later reconstructed in 3D. The dendritic shaft area and total spines area were calculated from the reconstructions and resulted in a F value for each branch (***Equation 2***). Scale bar is 5 μm.

studies when quantifying dendritic spines with the Golgi method, e.g., (***Feldman and Peters, 1979***) were not used in the present study as the fluorescent labeling and the high power reconstruction allowed the visualization of dendritic spines that protrude from the underside of dendrites. However, confocal stacks of images intrinsically result in a z-dimension distension; thus, a correction factor of 0.84 was applied to that dimension. This factor was calculated using a 4.2 μm TetraSpeck fluorescent microspheres (Molecular Probes) under the same parameters used for the acquisition of dendritic stacks. No optical deconvolution was used for spine reconstruction. See (***Benavides-Piccione et al., 2013***) for a detailed methodology of spine reconstruction.

For each individual dendritic spine, a particular threshold was selected to constitute a solid surface that exactly matched the contour of each dendritic spine. However, many times it was necessary to use several surfaces of different intensity thresholds to capture the complete morphology of a dendritic spine. In these cases, Imaris did not give a proper estimation of the spine area. Thus, spine 3D reconstructions had to be further processed in order to accurately calculate the surface area of the spines. We developed a new method for preprocessing the resulting meshes in order to join all the pieces into one solid mesh per spine. For each spine we made the following steps: (i) rasterize the meshes into a high-resolution 3D-mask; (ii) dilate and erode the foreground just the minimum amount of voxels V needed to join all the connected components into only one; (iii) reconstruct a mesh from the foreground using Marching Cubes (***Lorensen and Cline, 1987***) after applying a Gaussian Filter of V voxels. The resulting mesh is an accurate reconstruction of the spine from which the volume and the area can be accurately calculated because each spine represents a 3D-model with no visible differences from the originals. Thus, we concluded that the post-processed meshes are as accurate as the pre-processed ones. The implementation used in this dataset is based on VTK library (http://www.vtk.org/). A total of 7917 and 8345 spine membrane areas from the cingulate and temporal cortex, respectively, were included in this study.

## Electrophysiology (acute living slices)
### Voltage transients
Short (2 ms) subthreshold current pulses of varying amplitude and polarity were injected via whole-cell somatic current injection in six L2/3 human pyramidal cells and in four L2/3 mouse pyramidal cells. Brief current pulses were used in order to minimize the activation of voltage-dependent mechanisms. Each stimulus was repeated 50 times and the voltage response was averaged (***Figure 1***, black traces and ***Figure 1—figure supplement 2***). These experimental voltage transients were used as a target for fitting theoretical transients generated by a model of that same 3D reconstructed cell (color traces in ***Figure 1a,d*** and ***Figure 1—figure supplement 2b***). Optimization of cable parameters (specific membrane resistivity, $R_m$, and capacitance, $C_m$, and specific cytoplasm resistivity, $R_a$) was first performed by fitting model transients to the experimental voltage transient for 200 pA

**Table 2.** Various parameters for the five patients from which the eleven L2/3 pyramidal cells that were used in this study where taken. CLB, clobazam; CBZ, carbamazepine; FRI, frisium; LCS, lacosamide; LEV, levetiracetam; LTG, lamotrigine; MDZ, midazolam; MTS, mesiotemporal sclerosis; OXC, oxcarbazepine; ZGR, zonegran.

| Gender | Age (years) | Age at epilepsy onset (years) | Diagnosis | Seizure frequency (per month) | Antiepileptic drugs (pre-surgery) | Data used in figure |
|---|---|---|---|---|---|---|
| Male | 25 | 24 | Tumor | 8 | LEV, CBZ, LCS | *Figure 1*: a, c2, c3 |
| Female | 45 | 23 | MTS (meningitis) | 3 | CBZ, CLB, LTG | *Figure 1*: c1, c4, c5 |
| Female | 40 | 16 | MTS | 7 | ZGR, LTG, FRI, MDZ | *Figure 2* |
| Male | 43 | 6 | MTS | 7 | LEV, OXC | *Figure 2* |
| Male | 51 | 4 | unknown etiology | 60 | CBZ, FRI | *Figure 2* |

depolarization. The resultant models (fitted for only one transient) were than validated on different (depolarizing and hyperpolarizing) stimuli (*Figure 1b*). In one mouse neuron (left cell in *Figure 1—figure supplement 2a1*), we obtained transient response to only 100 pA hyperpolarization; the respective model was thus fitted this experimental transient. Throughout, recorded membrane potential was corrected for a calculated liquid junction potential of 16 mV.

## Nucleated patches

Nucleated outside-out patch recordings were made as described previously (*Gentet et al., 2000*; *de Kock et al., 2004*). The intracellular recording solutions (in mM) for nucleated patch recordings consisted of: 70 Cs-Gluconate, 70 CsCl, 10 HEPES, 0.5 NaGTP, 5 Mg2-ATP, 10 EGTA, 10 K-Phosphocreatine (pH 7.3, 290 mOsm). Recording pipettes were made of borosilicate glass and were coated with Sylgard. After fire-polishing the tips, pipette resistance was 2.5–4 MΩ. Series resistance was not compensated. Pipette capacitance was fully compensated while in the cell-attached configuration. Pyramidal neurons were targeted for recording with infrared differential interference contrast (IR-DIC) microscopy (*Figure 2*). After establishing whole-cell configuration, a nucleated patch was pulled from the cell body and lifted above the slice close to the surface of the bath to reduce pipette capacitance. Currents were recorded using a Multiclamp 700B amplifier (Molecular Devices, CA). Data was recorded at 250 kHz and low-pass-filtered offline at 50 kHz. A −5 mV pulse was applied from a membrane potential of −60 mV and 400–800 capacitive transients were recorded and averaged for analysis. At the end of the recording, the patch was ruptured and the open tip of the pipette was pressed against a small sylgard ball, resulting in a GΩ seal and a −5 mV pulse was applied (*Gentet et al., 2000*). Care was taken that the immersion depth of the pipette was maintained constant. To calculate the specific membrane capacitance of the nucleated patch, the procedures and formulas described in (*Gentet et al., 2000*) were used. The surface area for both human and mice nucleated patches was estimated using *Equation (1)*.

$$Surface\ area = (major\ axis + minor\ axis)^2 \left(\frac{\pi}{4}\right) \tag{1}$$

Surface areas for the human cells were 371.9 ± 131.5 μm$^2$ (n = 5) and for the mice 250.8 ± 31.1 μm$^2$ (n = 7). Nucleated patches with input resistances below 200 MΩ were excluded from analysis. Time constants were derived from single exponential fits of the traces recorded from the nucleated patches, as well as the traces from which the residual capacitive transient was subtracted from the capacitive transient recorded from the nucleated patch. Both analyses resulted in quantitatively similar estimates of the specific membrane capacitance in μF/cm$^2$ (human 0.47 ± 0.15 and 0.51 ± 0.11, mouse 1.03 ± 0.27 and 0.83 ± 0.34) and the between species comparisons were significantly different in both analyses.

## Patient history
Data available in *Table 2*.

## Models and simulations

Simulations were performed using NEURON 7.4 (*Carnevale and Hines, 2006*) running on grid of 40 Intel(R) Xeon(R) CPU E5-2670 with 16 cores per node (640 cores in total), running Redhat 6.6.

The modeled cells (in NEURON/Python code), can be found in the *Source code 1* and in ModelDB (https://senselab.med.yale.edu/ModelDB/showmodel.cshtml?model=195667).

### Incorporating dendritic spines globally into the modeled cell

The spine membrane area was incorporated globally into the 3D reconstructed dendritic model using the factor F,

$$F = \frac{dendritic\ membrane\ area + total\ spine\ area}{dendritic\ membrane\ area} \qquad (2)$$

The incorporation was achieved by multiplying $C_m$ by F and dividing $R_m$ by F as described previously (*Rapp et al., 1992*). This approximation is valid when current flows from dendrites to spines (e. g., for somatic current stimulus) because, in this case, spine head and spine base are essentially isopotential (*Jack et al., 1975*; *Segev and Rall, 1988*; *Rapp et al., 1992*). This holds even in the case of very large spine neck resistance values (not shown). However, when the current is from the spine to the dendrite (e.g., following the activation of a spiny synapse), the spine head membrane and the dendritic base are far from being isopotential. Then the above approximation is invalid and the stimulated spines must be modeled individually (*Figure 3*).

F in *Equation (2)* was computed using detailed data from (*Benavides-Piccione et al., 2013*) on human cingulate cortex and from unpublished data on L2/3 cells from the human temporal cortex (*Figure 4*). Spine and shaft areas were computed using reconstructions of 3D images from confocal microscopy, see *Figure 4* and (*Benavides-Piccione et al., 2013*). Samples from two human males (aged 40 and 85) were used in this study (see above). For the brain of the 85-year old man F values were as follows: Cingulate cortex: basal dendrites F = 1.81 ± 0.34, apical dendrites F = 1.78 ± 0.33; Temporal cortex: basal dendrites F = 1.89 ± 0.41, apical dendrites F = 1.87 ± 0.13. For the 40-year old: Cingulate cortex: basal dendrites F = 1.98 ± 0.38, apical dendrites F = 2.00 ± 0.28; Temporal cortex: basal dendrites F = 2.39 ± 0.63, apical dendrites F = 2.39 ± 0.27. Taking into account all the spines and the dendrites in the data, the average F value was 1.946 (*Figure 4*). For the rest of the work we have used *F* = 1.9 (implying that almost 50% of the dendritic membrane area in L2/3 neurons are in dendritic spines). The spine membrane area was incorporated using the F factor only in those segments that are at a distance of at least 60 µm from the soma, due to the very small density of spines in more proximal branches (*Benavides-Piccione et al., 2013*).

Due to lack of data of the spine area and density on L2/3 of the mouse temporal cortex, we assumed a F factor for the mouse models that was the same as that for the human neurons (F = 1.9). This value is in the range used in recent modeling studies for rodent L2/3 neurons in the whisker and somatosensory cortices (*Sarid et al., 2015*; *Palmer et al., 2014*). In our mouse models F was constant throughout the whole dendritic tree.

### Model for dendritic spines

Spines receiving synaptic inputs were modeled in full (*Figure 3*) using two compartments per spine. One for the spine neck and one for the spine head. The dimensions for these compartments were based on the data from the laboratory of Javier DeFelipe. The spine neck was modeled using a cylinder of length 1.35 µm and diameter of 0.25 µm, whereas the spine head was modeled as an isopotential compartment with a total area of 2.8 µm². The passive parameters ($C_m$, $R_m$, $R_a$) of the spine were similar to those of the dendrites. This spine model led to a spine neck resistivity of 50–80 MΩ.

### Fitting passive parameters using compartmental modeling

Six detailed compartmental models of the six 3D reconstructed layer 2/3 pyramidal neurons from human temporal cortex were fitted to match the experimental voltage transients of these cells (*Figure 1*). In order to fit the experimental transients, we optimized the values of the three key passive parameters: $C_m$, $R_a$, $R_m$. The optimization ran with the 'Multiple Run Fitter' tool in NEURON (*Carnevale and Hines, 2006*). Optimization was achieved by minimizing the root-mean-square deviation (RMSD) between the experimental data and the model response in a time window between 1–

100 ms following the succession of the brief depolarizing current pulse (*Rall et al., 1992*; *Major et al., 1993*; *Major et al., 1994*). The same method was also used for fitting the passive parameters for four L2/3 pyramidal cells from the mouse temporal cortex (*Figure 1—figure supplement 2*).

## Simulations with non-uniform $R_m$ and with Ih

To confirm that our findings also hold if we assume non-uniform cable properties, we ran different sets of simulations where we allowed non-uniform dendritic $R_m$. Optimizations were performed with five parameters: $C_m$, $R_a$, $R_m(soma)$, $R_m(end)$ and $d_{half}$ (*Equation (3)* below). $R_m$ values in the different compartments were calculated according to the following equation, as in (*Golding et al., 2005*):

$$R_m = R_m(end) + \frac{R_m(soma) - R_m(end)}{1 + exp[(dis - d_{half})/steep]}$$

(3)

where *dis* is the physical distance of the compartment from the soma and *steep* was assumed to be 50 μm, and the maximal dendritic distance in the six human cells was 1050 ± 178 μm.

In a different set of simulations, we optimized the passive parameters of the model under the assumption that Ih is expressed in these cells. We used (*Kole et al., 2006*) Ih model, where the somatic Ih density was chosen to attain one of the following values [0 (passive), 0.1, 0.2 mS/cm$^2$]. The basal dendrites had the same channel density as in the cell body, whereas the density of Ih in the apical tree increased exponentially with distance from soma according to the following equation:

$$Ih = Ih(soma) * \left(-0.8696 + 2.087 * e^{dis/323}\right)$$

(4)

## Statistical and systematic errors

Statistical errors in the best fit parameters were estimated by balanced resampling of the experimental voltage transients. Each trace was indexed between 1 to 50. To generate 100 synthetic, resampled data sets, these index sets were repeated 100 times each to form a matrix of size 100 × 50. Random permutations were applied to the matrix, such that each index still occurred 100 times in the matrix, but at random positions. Finally, the matrix was partitioned into 100 sublists of length 50, each representing an index set for a resampled average impulse response. For further details about this method see (*Roth and Häusser, 2001*).

The influence of possible systematic errors on best-fit model parameters and predictions was investigated in a similar way (*Roth and Häusser, 2001*). The five most likely independent sources of systematic errors were considered. Error variables were assumed to be normally distributed around the mean given by the original morphology, with standard deviations estimated according to the expected experimental uncertainties (*Table 3*).

## Synaptic inputs

The synaptic input in *Figure 3* was based on both AMPAR and NMDAR mediated currents that were simulated as

$$I_{syn} = g_{syn}(t, V) * (V - E_{syn})$$

(5)

**Table 3.** Sources of systematic errors.

| Error variable | Estimated S.D. |
| --- | --- |
| Scale factor for lengths in the morphological reconstruction | 0.05 |
| Additive error in reconstructed diameters (μm) | 0.3 |
| Multiplicative error in reconstructed diameters | 0.1 |
| Error in spine scale factor, F | 0.4 |
| Error in the start point of high density spines (μm) | 30 |

where $g_{syn}$ is the synaptic conductance change, and $E_{syn}$ is the reversal potential for the synaptic current. $E_{syn}$ was 0 mV for both the AMPAR and the NMDAR mediated currents.

The synaptic conductance was modeled for both AMPA and the NMDA components, using two-state kinetic synaptic models – with rise time ($\tau_{rise}$) and decay time ($\tau_{decay}$) constants:

$$g_{syn}(t, V) = B * g_{\max} * N * \left(\exp(-t/\tau_{decay}) - \exp(-t/\tau_{rise})\right) \tag{6}$$

Here $g_{\max}$ is the peak synaptic conductance and $N$ is a normalization factor given by

$$N = \frac{1}{\exp\left(-t_{peak}/\tau_{decay}\right) - \exp\left(-t_{peak}/\tau_{rise}\right)} \tag{7}$$

and $t_{peak}$ (time of the peak conductance) is calculated as:

$$t_{peak} = \frac{\tau_{rise} * \tau_{decay}}{\tau_{decay} - \tau_{rise}} * \log\left(\frac{\tau_{decay}}{\tau_{rise}}\right) \tag{8}$$

NMDA conductance is voltage-dependent. In this work, B was defined using the equation as in (*Jahr and Stevens, 1990*)

$$B = \frac{1}{1 + \exp(-\gamma * V) * [Mg^{2+}] * n} \tag{9}$$

*Table 4* summarizes the different parameters for the two synaptic components; AMPA and NMDA parameters were, as in recent research on rodents (*Sarid et al., 2007*; *Jahr and Stevens, 1990*; *Rhodes, 2006*; *Larkum et al., 2009*).

## Axon

In *Figure 3*, A very long unmyelinated axon was added to the model of the cell in *Figure 1a* in order to assess the impact of the specific capacitance on the conductance velocity of the spike along the axon. The modeled axon was 6 mm long with a diameter of 1 μm; its passive membrane properties were as in *Figure 1* ($R_a$ = 203 Ωcm, $R_m$ = 38,907 Ωcm$^2$ and $C_m$ = 0.45 μF/cm$^2$).

## Active ion channels

To the detailed 3D model of HL2/3 PC we added a simplified excitable model for the generation of Na$^+$ spikes at the soma. We aimed at preserving both the current threshold (~300 pA) and the voltage threshold (~ +20 mV above resting potential) for spike initiation as found in in vitro experiments in these cells (*Verhoog et al., 2013*; *Testa-Silva et al., 2014*). To achieve that we used Hodgkin-Huxley formalism (*Mainen and Sejnowski, 1996*) with maximal Na$^+$ and K$^+$ conductances at the soma of 8000 pS/μm$^2$ and 3200 pS/μm$^2$, respectively. The V1/2 for the sodium activation curve was shifted by −8 mV so that the modeled cell generated a spike at ~ −65 mV, as in the experiments. A long, unmyelinated axon, (see above) was coupled to the soma, consisting of maximal Na$^+$ and K$^+$ conductances of 200 pS/μm$^2$ and 100 pS/μm$^2$, respectively.

**Table 4.** Synaptic properties for *Figure 3*.

|  | AMPA | NMDA |
| --- | --- | --- |
| $\tau_{rise}(ms)$ | 0.3 | 3 |
| $\tau_{decay}(ms)$ | 1.8 | 70 |
| $g_{\max}$ (nS) | 0.7 | 1.4 |
| $\gamma\left(\frac{1}{mV}\right)$ | - | 0.08 |
| $n\left(\frac{1}{mM}\right)$ | - | 1/3.57 |
| $[Mg^{2+}]$ | - | 1 mM |

## Number of spiny synapses per somatic spike

Dendritic spines with AMPA and NMDA synapses (see above), were randomly distributed over the dendrites of the active model (*Figure 3b*). The synapses were activated synchronously and the resultant somatic voltage was recorded. We counted somatic spikes when the voltage crossed a threshold of 0 mV. Each experiment (different number of synapses) was repeated 1000 times with different seeds.

## Acknowledgements

HDM received funding for this work from the Netherlands Organization for Scientific Research (NWO; 917.76.360, 912.06.148 and a VICI grant), ERC StG 'BrainSignals', the Dutch Fund for Economic Structure Reinforcement (FES, 0908 'NeuroBasic PharmaPhenomics project'), EU 7th Framework Programmes (HEALTH-F2-2009–242167 'SynSys' and grant agreement no. 604102 'Human Brain Project'). Part of this project was supported by Hersenstichting Nederland (grant HSN 2010(1)-09 to CPJdK). JDF was supported by the Spanish Ministry of Economy and Competitiveness through the Cajal Blue Brain (C080020-09; the Spanish partner of the Blue Brain initiative from EPFL), and by the European Union's Seventh Framework Programme (FP7/2007–2013) under grant agreement no. 604102 (Human Brain Project). IS was supported by grant agreement no. 604102 'Human Brain Project' and by a grant from the Gatsby Charitable Foundation.

## Additional information

### Funding

| Funder | Grant reference number | Author |
| --- | --- | --- |
| Ministerio de Economía y Competitividad | the Cajal Blue Brain (C080020-09; the Spanish partner of the Blue Brain initiative from EPFL) | Javier DeFelipe |
| European Union's Seventh Framework Programme | (FP7/2007-2013) under grant agreement mo. 604102 (Human Brain Project) | Javier DeFelipe |
| Hersenstichting Nederland | grant HSN 2010(1)-09 | Christiaan PJ de Kock |
| Nederlandse Organisatie voor Wetenschappelijk Onderzoek | NWO; 917.76.360 | Huibert D Mansvelder |
| European Research Council | BrainSignals 281443 | Huibert D Mansvelder |
| Nederlandse Organisatie voor Wetenschappelijk Onderzoek | NWO; 912.06.148 | Huibert D Mansvelder |
| Nederlandse Organisatie voor Wetenschappelijk Onderzoek | NWO; VICI grant | Huibert D Mansvelder |
| Human Brain Project | grant agreement no. 604102 | Idan Segev |
| Gatsby Charitable Foundation | | Idan Segev |

The funders had no role in study design, data collection and interpretation, or the decision to submit the work for publication.

### Author contributions

GE, Designed the research, Performed the simulations, Analyzed the data, Wrote the paper; MBV, GT-S, JCL, HDM, Did the electrophysiology; YD, Analyzed the data; RB-P, JM, JD, Reconstructed human spines; CPJdK, Reconstructed the human cells; IS, Designed the research, Wrote the paper, Supervised the research

### Author ORCIDs

Guy Eyal, http://orcid.org/0000-0002-9537-5571
Idan Segev, http://orcid.org/0000-0001-7279-9630

**Ethics**

Human subjects: All procedures on human tissue were performed with the approval of the Medical Ethical Committee (METc) of the VU University Medical Centre (VUmc), with written informed consent by patients involved to use brain tissue removed for treatment of their disease for scientific research, and in accordance with Dutch license procedures and the declaration of Helsinki (VUmc METc approval 'kenmerk 2012/362').

Animal experimentation: All animal experimental procedures were approved by the VU University's Animal Experimentation Ethics Committee and were in accordance with institutional and Dutch license procedures (approved protocol INF09-02A1V1).

## Additional files

**Supplementary files**

• Source code 1. NEURON code for the human models in this work.

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
