## [Decision Letter]

Thank you for submitting your article "Unique Membrane Properties and Enhanced Signal Processing in Human Neocortical Neurons" for consideration by *eLife*. Your article has been reviewed by two peer reviewers, and the evaluation has been overseen by Michael Häusser as the Reviewing Editor and Eve Marder as the Senior Editor. The following individual involved in review of your submission has agreed to reveal their identity: Nelson Spruston (Reviewer).

The reviewers have discussed the reviews with one another and the Reviewing Editor has drafted this decision to help you prepare a revised submission.

Summary:

This manuscript examines the cable and specific membrane properties of human neocortical neurons in slices dissected from patients. To address this question, the authors use electrophysiological recording, cable modelling, and simulations. The main finding is that the specific capacitance in human cells is lower than previously reported in rodents, as convergently shown by two different approaches (cable modeling and nucleated patch recording). Furthermore, they show that a low Cm is advantageous for both efficacy and speed of signal propagation in pyramidal cells. If the difference in Cm is indeed true, this would clearly be an important and interesting finding that surely deserves publication in *eLife*. However, several issues need to be addressed before the conclusions can be considered secure. Most importantly, alternative interpretations of the data are not carefully considered. Furthermore, the case for a difference in Cm rests on only four recordings from mouse neurons.

Essential revisions:

1) Short current pulses in current clamp: why were these performed only in human neurons and not in mouse neurons? The decay of voltage transients in response to current injection is a highly indirect measure of Cm. Thus, it would be valuable to identify the fundamental difference in the voltage responses observed in human and mouse neurons, thus providing a direct indication of the feature of the data that is indicative of a different Cm. If the authors have short current pulse data from mouse neurons, they should therefore provide it, particularly since this would offer two parallel lines of evidence that illustrates the difference between human and mouse neurons, which would be more convincing than just the result from nucleated patches.

2) Nucleated patch experiments: the number of recordings is very low. In particular, n=4 for mouse neurons is very low by the standards of the field, especially since obtaining recordings from mouse neurons is easier than for human neurons. We strongly encourage the authors to add additional datapoints to the mouse nucleated patch dataset.

3) Modeling of human neurons: were these the same neurons from which the recordings were obtained? If not, it is possible that the morphologies used were different enough to introduce significant error in the estimation of Cm. (Note: this is an example of a general problem with the paper, which is that it is not well written.).

4) Modeling of human neurons: Fits of the voltage transients are based on estimates of Ra, Rm, and Cm. To make a more convincing case that low values of Cm are necessary to achieve good fits, please provide a plot (or plots) of RMS error for different combinations of these three values. See also additional comments below about series-resistance and capacitance compensation.

5) The Ra obtained in the present study seems unrealistically high. Two-electrode recording (separating voltage-recording from current-feeding electrode) has revealed Ra values of 115 – 190 Ohm cm (Roth and Häusser, 2001; Schmidt-Hieber et al., 2007; Nörenberg et al., 2010). Clearly, the best way to address this discrepancy would be to use the two-electrode approach for human neurons. If this is not possible, the limitations of the present study should be clearly stated. Additionally, the value of Ra will depend on the intracellular solution. However, the composition of the internal solution for the whole-cell recordings is not even mentioned in the paper (it is most likely different from the Cs+ solution used for nucleated patch recording).

6) There is evidence for non-uniformity of Rm in the literature (Stuart and Spruston, 1998; Nörenberg et al., 2010). The authors should test non-uniform models to corroborate the conclusion of low Cm under these conditions.

7) The authors need to obtain confidence intervals of the parameter estimates. One possibility might be to use bootstrap analysis, as previously suggested by Roth and Häusser, 2001.

8) Cable modeling is well known to be sensitive to even subtle nonlinearities. A particular problem is Ih. Is Ih expressed in the human neurons? Is there a sag during hyperpolarizing currents mediated by Ih? Did the authors make any attempt to block Ih?

9) The accuracy of the Cm estimate stands and falls with the reliability of the spine correction. However, the spine correction factors are not convincing. Wouldn't it be the best to directly count spines in the recorded cells rather than in postmortem tissue? How did the authors correct for hidden spines (branching out in z direction) or for spines below the LM resolution limit? Finally, without being pedantic, the mean F from the given data is 2.0, rather than 1.9. We agree that a change will make the Cm even smaller, but in any case, this highlights potential systematic errors.

10) The estimation of the surface area of the nucleated patches is not convincing. The shape of the nucleated patches is probably best approximated by an ellipsoid. The exact formula for the surface area of spheroids or triaxial ellipsoids is quite complicated, so the simple equation 4 of Gentet et al., 2000 (which the authors apparently used; see subsection “Nucleated Patches”) is an approximation of an approximation. Also, the authors don't state which of the "formulas" of the Gentet paper they used. Finally, numbers for the surface area of the nucleated patches need to be given in the present paper.

11) Unfortunately, the authors fail to address the mechanisms underlying the low Cm in human pyramidal cells. Is it a difference in the relative dielectric constants or the geometric properties of the lipids (i.e. the length of acyl side chains)? Or does the protein content influence the relative dielectric constant of the membrane (and thereby Cm)? At the very least, these aspects have to be better discussed.

---

## [Author Response]

[…]

*Essential revisions:*

*1) Short current pulses in current clamp: why were these performed only in human neurons and not in mouse neurons? The decay of voltage transients in response to current injection is a highly indirect measure of Cm. Thus, it would be valuable to identify the fundamental difference in the voltage responses observed in human and mouse neurons, thus providing a direct indication of the feature of the data that is indicative of a different Cm. If the authors have short current pulse data from mouse neurons, they should therefore provide it, particularly since this would offer two parallel lines of evidence that illustrates the difference between human and mouse neurons, which would be more convincing than just the result from nucleated patches.*

Following the reviewers’ request, we repeated the same experimental and theoretical protocols as in human neurons also on mouse L2/3 PCs, from the temporal cortex (n = 4). In contrast to human neurons, we found in mice, *C_m_* of 1.09 ± 0.36 µF/cm, mean ± S.D (Figure 1—figure supplement 2). This value agrees with that found in previous studies (Major et al. 1994; Gentet et al. 2000; Roth & Häusser 2001; Nörenberg et al. 2010; Szoboszlay et al. 2016) and is significantly larger compared with what we had found in human L2/3 neurons (students t-test without the assumption of equal variance, p-val = 0.0408, non-parametric Wilcoxon rank-sum test, p-val = 0.0095).

*2) Nucleated patch experiments: the number of recordings is very low. In particular, n=4 for mouse neurons is very low by the standards of the field, especially since obtaining recordings from mouse neurons is easier than for human neurons. We strongly encourage the authors to add additional datapoints to the mouse nucleated patch dataset.*

We wish to emphasize that prime objective of these recordings was to test whether the nucleated patch recordings of adult human pyramidal neurons would confirm the model estimates of a specific membrane capacitance of 0.5 µF/cm^2^or not. The recordings from the mouse nucleated patches confirmed well-established existing data and values reported in the literature for mouse neurons (Gentet et al., 2000; Szoboszlay et al., Neuron, 2016) n=4 may not be a high number, but a same number of nucleated patches was also used in other studies for similar type of experiments, Szoboszlay et al., (Neuron 2016), in which nucleated patch measurements on mouse cerebellar neurons resulted in a value of 1 µF/cm^2^based on n=4 recordings. Nevertheless, we added three new recordings on mouse nucleated patches to the revised manuscript (now n=7, see revised Figure 2). Furthermore, to ensure that the specific membrane capacitance estimates based on the nucleated patch were robust, we derived time constants from single exponential fits of the traces recorded from the nucleated patches, as well as the traces from which the residual capacitive transient was subtracted from the capacitive transient recorded from the nucleated patch. Both analyses resulted in quantitatively similar estimates of the specific membrane capacitance (human 0.47 ± 0.15 and 0.51 ± 0.11, mouse 1.03 ± 0.27 and 0.83 ± 0.34) and the between species comparisons were significantly different in both analyses.

The combined *C_m_*values obtained for human neurons in both methods, the model fittings (Figure 1) and the nucleated patches (Figure 2), are significantly different from the *C_m_*values obtained for the mouse neurons (Figure 2, human n=11, mouse n=11, students t-test following the KS test for normality, p-val = 0.0021, non-parametric Mann Whitney, p-val = 0.006).

*3) Modeling of human neurons: were these the same neurons from which the recordings were obtained? If not, it is possible that the morphologies used were different enough to introduce significant error in the estimation of Cm. (Note: this is an example of a general problem with the paper, which is that it is not well written.).*

Yes – transients were obtained from the same 3D reconstructed morphology for all 6 L2/3 cells (Figure 1). This is now better clarified in the following sentences:

“In Figure 1, fifty brief depolarizing current pulses (200 pA, 2 ms each) were injected into the soma of the HL2/3 PC shown at the right; the resultant averaged voltage transient is depicted by the black trace. […] The 3D anatomy plus physiology were combined for reconstructing a detailed model for that cell.”

*4) Modeling of human neurons: Fits of the voltage transients are based on estimates of Ra, Rm, and Cm. To make a more convincing case that low values of Cm are necessary to achieve good fits, please provide a plot (or plots) of RMS error for different combinations of these three values. See also additional comments below about series-resistance and capacitance compensation.*

We thank the reviewers for this good suggestion. Figure 1—figure supplement 1 was added to the paper.

This supplementary figure is now referred to in the main text as follows:

“To validate that low C_m_ values are indeed necessary to achieve good fits between the theoretical and the experimental transients, we plotted (Figure 1—figure supplement 1) the root mean square deviation (RMSD) for different combinations of the three passive parameters (C_m_, R_a_, R_m_) that impact the transients. […] We found that even with large systematic errors in morphological parameters, the best fit still yields low C_m_ value in human L2/3 pyramidal neurons (see Table 1).”

*5) The Ra obtained in the present study seems unrealistically high. Two-electrode recording (separating voltage-recording from current-feeding electrode) has revealed Ra values of 115 – 190 Ohm cm (Roth and Häusser, 2001; Schmidt-Hieber et al., 2007; Nörenberg et al., 2010). Clearly, the best way to address this discrepancy would be to use the two-electrode approach for human neurons. If this is not possible, the limitations of the present study should be clearly stated. Additionally, the value of Ra will depend on the intracellular solution. However, the composition of the internal solution for the whole-cell recordings is not even mentioned in the paper (it is most likely different from the Cs+ solution used for nucleated patch recording).*

We unfortunately cannot perform two-electrode recordings in human neurons. But in the new Figure 1—figure supplement 1 we found that, assuming *R_a_* ~ 150 Ωcm, the fit is slightly worse compared with the case of higher *R_a_* (12% ± 10 increase in RMSD comparing with the case of free parameters, *R_a_* = 268.5 ± 30.0, n=6). Note that similar high values of *R_a_* were also obtained by (Major et al. 1994), with a single electrode.

However, as shown in Figure 1—figure supplement 1, the influence of *R_a_* on the fit is modest (shallower) compared with the influence of *C_m_*.

As shown in Figure 5, the high *R_a_* value is required only for the very first 2 ms of the transient. Thus, we agree with the referee that our estimation for the value of *R_a_* is less accurate than for *C_m_*. This point is discussed in the main text (see point #4 above).

Author response image 1.Impact of *R_a_* value on the quality of the fit of the theoretical versus the experimental transients.(**a**) Experimental (black trace) and best theoretical fit (green trace) as in Figure 1. (**b**) Zoom in on the green square in **a**. (**c**–**e**) Same as in **b**, but with *R_a_*constrained to 100, 200 and 300 Ω•cm correspondingly. Note the deviation of the fit from the experimental trace in the first couple of ms in c and e. Parameters resulted for the various optimization:c*. R_a_* = 100 Ω•cm (constrained), *C_m_*= 0.42 μF/cm^2^, *R_m_*= 41,315 Ω•cm^2^; d. *R_a_* = 200 Ω•cm (constrained), *C_m_*= 0.46 μF/cm^2^, *R_m_*= 38,299 Ω•cm^2^ ;e. *R_a_* = 300 Ω•cm (constrained), *C_m_*= 0.50 μF/cm^2^, *R_m_*= 36,372 Ω•cm^2^.**DOI:**
http://dx.doi.org/10.7554/eLife.16553.015

The internal solutions used in the human and mouse neuron recordings that were used for the model simulations are listed in Verhoog et al., 2013 and Testa-Silva et al., 2014. For clarity, we have added the composition of the solutions to the Methods section.

*6) There is evidence for non-uniformity of Rm in the literature (Stuart and Spruston, 1998; Nörenberg et al., 2010). The authors should test non-uniform models to corroborate the conclusion of low Cm under these conditions.*

We thank the referee for this point. Below, we show that, in general, the small *C_m_* is required in order to fit the first few milliseconds of the transient, no matter what are the active/passive ion currents in the model. This is also true for the case of non-uniform dendritic *R_m_*, as well as for the case where Ih is expressed L2/3 human neurons (see point #8 below).

We ran different sets of simulations with non-uniform *R_m_*. Optimizations were performed with five parameters: *C_m_, R_a_, R_m_* (soma), *R_m_* (end) and dhalf. *R_m_* values in the different compartments were calculated according to the following equation (as in Goldin et al., 2005):

Rm=Rm(end)+ Rm(soma)−Rm(end)1+e(dis−dhalf)/steep where *dis* is the physical distance of the compartment from the soma and *steep* was assumed to be 50 µm. The results are displayed below:

Author response image 2.Summary of model parameters for the case of non-uniform *R_m_*.Model transients were fitted to the experimental voltage transients as in Figure 1. Colors match the colors in Figure 1. (**a**) Comparison of *C_m_*value for the six human models for the case of uniform *R_m_*(U) and the case of non-uniform *R_m_*(N). Bars show the mean value. (**b**) Root mean square distance between the model responses and the experimental transients. (**c**) *R_a_*values for the two cases. (**d**) *R_m_*(soma) versus *R_m_*(end) for the six models, when *R_m_* was assumed to be non-uniform. (**e**) Normalized dhalf for the six models; the maximal dendritic distance in the cells was 1050 ± 178 µm.**DOI:**
http://dx.doi.org/10.7554/eLife.16553.016

As you can see, the algorithm converged to different solutions for different cells; however, in all cells the estimated *C_m_* and *R_a_* values were close to those obtained in the case of uniform *R_m_*, and, thus, the conclusion regarding the unique *C_m_* value in human neurons remains unchanged.

Consequently, we added the following paragraph:

“Could it be that our uniform passive model used in Figure 1 is too simplistic? […] In all six cases the estimated C_m_ for the optimal non-uniform case was similar to the value obtained when R_m_ was uniform (not shown).”

*7) The authors need to obtain confidence intervals of the parameter estimates. One possibility might be to use bootstrap analysis, as previously suggested by Roth and Häusser, 2001.*

To address point 7, we repeated both the bootstrap analysis as well as the systematic error analysis as suggested by Hausser and Roth (2001). Consequently, the paragraph below, and the accompanied tables, were added to the revised version of this work.

“We further examined whether possible statistical and systematic errors in the data might effected our estimates of C_m_, R_m_ and R_a_ by performing error analysis as in (Roth & Häusser 2001). We found that even with large systematic errors in morphological parameters, the best fit still yields low C_m_ value in human L2/3 pyramidal neurons (see Table 1).”

*8) Cable modeling is well known to be sensitive to even subtle nonlinearities. A particular problem is Ih. Is Ih expressed in the human neurons? Is there a sag during hyperpolarizing currents mediated by Ih? Did the authors make any attempt to block Ih?*

This is indeed a very good point, and we thank the reviewers. Unfortunately, we did not block Ih in the human experiments. Indeed, L2/3 pyramidal cells in human might express Ih as a clear sag in both depolarizing and hyperpolarizing voltage traces were found in these cells (see Figure 1 in Verhoog, 2013). However, our analysis below shows that even in the presence of Ih, a low *C_m_* around 0.5 is still required to reconcile between the model and the experimental results.

In order to explore the effect of Ih on the model estimates of *C_m_*, we added Ih to L2/3 neuron models using Kole et al., 2006 model for Ih. In this model, Ih density increases exponentially from soma to the apical dendrites (see equation below). As in the passive case, we attempted to fit the transients generated by the model (with Ih) to the experimental transients; this was performed for different densities of somatic (and therefore the dendritic) Ih.

We found that the larger the Ih density, the larger was the estimated *C_m_* value that best fitted the whole experimental transient (from ~0.45 μF/cm^2^without Ih, Figure 1—figure supplement 3, to ~0.76 μF/cm^2^(in Figure 1—figure supplement 3) with soma Ih density of 0.2 mS/cm, as used by Kole et al., 2006). However, the quality of this fit was reduced with increasing Ih density, as demonstrated by the increase in the respective RMSD values (Figure 1—figure supplement 3). This reduction in the quality of fit was particularly evident at short times (Figure 1—figure supplement 3).

The increase in the estimated *C_m_* in the presence of Ih is expected because Ih effectively decreases the membrane resistivity and thus the effective membrane time constant (by adding a conductance in parallel to the passive conductance). Consequently, in order to match the “tail” of the theoretical transient (that is governed by the effective time constant) to the experimental tail, a larger modeled *C_m_* is required. However, at short times, the voltage transient is dominated by the capacitive current (and not by the ionic currents, e.g., Ih); at these short times, a good match between model and experiments could be only obtained with *C_m_* ~ 0.5 (Figure 1—figure supplement 3). But now the theoretical and experimental transients diverge significantly at later times; for Ih density of 0.2 mS/cm^2^ the theoretical curve (unlike the experimental curve) even exhibits an undershoot (Figure 1—figure supplement 3). We conclude that in order to explain the experimental transients in human L2/3 neurons *C_m_* should be ~ 0.5 otherwise we cannot explain the first few milliseconds of the experimental transients, even with active currents (Ih) presence in the model. Furthermore, if Ih is indeed expressed in human L2/3 neurons, then it is slow compared with that in rat L5 pyramidal cells (Kole et al. 2006). This is because, as shown in Figure 1—figure supplement 3), if fast Ih is expressed in human L2/3 neurons it was not possible to find a satisfactory match between the theoretical and experimental transients. Figure 1—figure supplement 3 was added to the paper.

Consequently, in the text we added the following paragraph:

“In the next set of simulations Ih was added to the model; the density of the Ih current increased with distance from soma as in (Kole et al. 2006), see Methods. […] We therefore conclude that in order to explain the experimental transients in human L2/3 neurons, C_m_ should be ~ 0.5 μF/cm^2^.”

*9) The accuracy of the Cm estimate stands and falls with the reliability of the spine correction. However, the spine correction factors are not convincing. Wouldn't it be the best to directly count spines in the recorded cells rather than in postmortem tissue? How did the authors correct for hidden spines (branching out in z direction) or for spines below the LM resolution limit? Finally, without being pedantic, the mean F from the given data is 2.0, rather than 1.9. We agree that a change will make the Cm even smaller, but in any case, this highlights potential systematic errors.*

This is indeed a key point as our estimation for *C_m_* critically depends on the estimated spine surface area. We initially did try to estimate the spine counts in the fresh tissue, but it became clear that we seriously underestimated their density due to hidden spines and thus we decided to use the best method available today at the LM level (using confocal microscopy); the lab. of Javier DeFelipe already has a large number of 3D reconstructed spines (a total of 7917 and 8345 spine membrane areas from the cingulate and temporal cortex, respectively, was included in this study). Some of these data has already been published (data on cingulate cortex; Benavides-Piccione et al., 2013), whereas the data on temporal cortex was not yet published.

Our method provided spine density which was significantly larger than the estimates obtained from the fresh tissue. Still, it could be that we miss spines (e.g., as compared to EM methods, see Lichtman et al., 2015). If this was the case, then our *C_m_* estimates would be even smaller than 0.5.

Following the reviewers comment we expanded the Methods section to explain in detail how we estimate the spine area; we also added a new figure (Figure 4) as was asked by one of the referees, showing three examples for dendritic spines in human L2/3 pyramidal cells (and the corresponding F value).

Note that taking all the spines and the dendrites in our data, the average F value was 1.946 (which was rounded to 1.9). This is now better explained in the Methods section.

The Methods section refereeing to dendritic spines now reads as below:

“Dendritic spines and dendritic branches were imaged using a Leica TCS 4D confocal scanning laser attached to a Leitz DMIRB fluorescence microscope. […] The implementation used in this dataset is based on VTK (http://www.vtk.org/) library. A total of 7917 and 8345 spine membrane areas from the cingulate and temporal cortex, respectively, were included in this study.”

*10) The estimation of the surface area of the nucleated patches is not convincing. The shape of the nucleated patches is probably best approximated by an ellipsoid. The exact formula for the surface area of spheroids or triaxial ellipsoids is quite complicated, so the simple equation 4 of Gentet et al., 2000 (which the authors apparently used; see subsection “Nucleated Patches”) is an approximation of an approximation. Also, the authors don't state which of the "formulas" of the Gentet paper they used. Finally, numbers for the surface area of the nucleated patches need to be given in the present paper.*

To estimate the surface area of the nucleated patches we used equation (4) from Getnet et al., (2000). It could be that this is approximation of approximation as the referee noted; however, we have used the exact same method for both human and mice neurons. Note that this same method was also used in a recent paper in Neuron (Szoboszlay et al., 2016) for estimating the *C_m_* value of Golgi cells. We now added the surface areas for the human and mouse nucleated patches together with the corresponding *C_m_* values. For human nucleated patches the surface area (in µm^2^) and *C_m_* (in μF/cm^2^) respectively were: 202, 0.38; 570, 0.48; 368, 0.44; 380, 0.59; 340, 0.65 and for the mouse: 232, 0.88; 209, 0.44; 287, 0.94; 222, 1.28; 252, 0.33; 269, 1.13; 285, 0.85. No correlation between the surface area and the calculated *C_m_* was found.

The following sentence is now added to the Methods:

“To calculate the specific membrane capacitance of the nucleated patch, the procedures and formulas described in (Gentet et al. 2000) were used. The surface areas for both human and mice were estimated using equation (1).(1)Surface area=(major axis+minor axis)2(π4)

Surface areas for the human cells were 371.9 ± 131.5 µm^2^ (n=5) and for the mice 250.8 ± 31.1 µm^2^ (n=7).”

*11) Unfortunately, the authors fail to address the mechanisms underlying the low Cm in human pyramidal cells. Is it a difference in the relative dielectric constants or the geometric properties of the lipids (i.e. the length of acyl side chains)? Or does the protein content influence the relative dielectric constant of the membrane (and thereby Cm)? At the very least, these aspects have to be better discussed.*

We thank the reviewers for this comment. Indeed, a better understanding of the mechanism underlying our result would strengthen the conclusions. However, we don’t know yet what the explanation is for the low *C_m_* value in human pyramidal neurons. We did measure the thickness of human vs. mouse membrane using high resolution EM, but both have the same (5nM) thickness (not shown). So the difference we found must be the result of the composition of the membrane bilayer; this should be studied by experts in membrane biophysics which is beyond the scope of our teams. However, following the reviewers’ comment we have now elaborated in the Discussion on what is known about membranes with respect to their capacitance.

*“*What could be the reason for the low C_m_ as found in the present work? The specific capacitance is determined by the dielectric constant of the material composing the membrane and by the membrane thickness. [… This assertion requires direct examination of the human membranes at the molecular level, which is beyond the scope of the present study.”